# Effect of the Lifestyle, Exercise, and Nutrition (LEAN) Study on Long-Term Weight Loss Maintenance in Women with Breast Cancer

**DOI:** 10.3390/nu13093265

**Published:** 2021-09-18

**Authors:** Alexa Lisevick, Brenda Cartmel, Maura Harrigan, Fangyong Li, Tara Sanft, Miklos Fogarasi, Melinda L. Irwin, Leah M. Ferrucci

**Affiliations:** 1Frank H. Netter MD School of Medicine, Quinnipiac University, Hamden, CT 06518, USA; miklos.fogarasi@quinnipiac.edu; 2Yale School of Public Health, Yale University, New Haven, CT 06510, USA; brenda.cartmel@yale.edu (B.C.); maura.harrigan@yale.edu (M.H.); fang-yong.li@yale.edu (F.L.); melinda.irwin@yale.edu (M.L.I.); leah.ferrucci@yale.edu (L.M.F.); 3Yale Cancer Center, New Haven, CT 06510, USA; tara.sanft@yale.edu; 4Yale School of Medicine, Yale University, New Haven, CT 06510, USA

**Keywords:** breast cancer, survivorship, weight loss maintenance, lifestyle intervention

## Abstract

Lifestyle interventions among breast cancer survivors with obesity have demonstrated successful short-term weight loss, but data on long-term weight maintenance are limited. We evaluated long-term weight loss maintenance in 100 breast cancer survivors with overweight/obesity in the efficacious six-month Lifestyle, Exercise, and Nutrition (LEAN) Study (intervention = 67; usual care = 33). Measured baseline and six-month weights were available for 92 women. Long-term weight data were obtained from electronic health records. We assessed weight trajectories between study completion (2012–2013) and July 2019 using growth curve analyses. Over up to eight years (mean = 5.9, SD = 1.9) of post-intervention follow-up, both the intervention (*n* = 60) and usual care (*n* = 32) groups declined in body weight. Controlling for body weight at study completion, the yearly weight loss rate in the intervention and usual care groups was –0.20 kg (−0.2%/year) (95% CI: 0.06, 0.33, *p* = 0.004) and −0.32 kg (−0.4%/year) (95% CI: 0.12, 0.53, *p* = 0.002), respectively; mean weight change did not differ between groups (*p* = 0.31). It was encouraging that both groups maintained their original intervention period weight loss (6% intervention, 2% usual care) and had modest weight loss during long-term follow-up. Breast cancer survivors in the LEAN Study, regardless of randomization, avoided long-term weight gain following study completion.

## 1. Introduction

In 2021, there will be an estimated 284,200 newly diagnosed cases of breast cancer and 44,130 breast cancer deaths in the United States, representing roughly 15% of all new cancer cases and 7% of all cancer deaths, respectively [1]. Currently, the five-year survival rate for breast cancer is 90% for all stages combined; nonetheless, mortality declines have slowed in recent years [1].

Obesity, defined as a body mass index (BMI) ≥30 kg/m^2^, in the setting of breast cancer survivorship has been associated with increased risk of recurrence, therapy-related morbidity [2], poorer overall and breast cancer-specific survival [3,4,5], as well as reduced quality of life [2]. The molecular mechanisms linking obesity and breast cancer biology are not entirely understood; however, it is suggested that hormones, adipocytokines, inflammatory cytokines, and reactive oxygen species play important roles [6]. Obesity, weight gain, and physical inactivity during or following cancer treatment are highly prevalent in breast cancer survivors [2,7,8]. An analysis of the National Health Interview Survey found the prevalence of obesity increased more rapidly among cancer survivors, compared to the general population, from 1997 to 2014 [9]. The annual increase in obesity prevalence of 3.0% among breast cancer survivors was one of the highest rates of increasing obesity burden among all cancer survivors [9]. Data from the most recent years in this national sample indicated approximately 30–35% of breast cancer survivors were obese [9].

Although weight gain amongst adult women as they age is common [10,11], weight gain among breast cancer survivors may start during treatment and continue months to years after diagnosis [12]. Furthermore, it appears that women who are normal weight at diagnosis more commonly experience post-diagnosis weight gain than women with overweight or obesity at diagnosis [13]. Importantly, in comparison to women who maintain their weight following diagnosis, those that experience weight gain have increased all-cause mortality, especially when weight gain is 10% or higher [14]. Thus, there is a growing emphasis on finding efficacious interventions focused on preventing weight gain or promoting weight loss among breast cancer survivors with overweight or obesity (BMI > 25 kg/m^2^) [2,15,16].

Lifestyle guidelines for breast cancer survivors highlight the importance of a healthy body weight with a focus on physical activity and diet [17]. The American Cancer Society and American Society of Clinical Oncology 2016 breast cancer survivorship care guidelines recommend physicians counsel survivors about consuming a diet high in vegetables, fruits, whole grains, and legumes, and low in saturated fats, as well as limiting alcohol intake. Further, recommendations for survivors include avoiding inactivity, completing at least 150 min of moderate or 75 min of vigorous aerobic exercise per week, and should include strength training exercises at least two days per week [18]. Some breast cancer survivors may have difficulty meeting these recommendations because of fatigue and therapy-related side effects, which may limit physical activity and achieving dietary goals. Data suggest only 18% and 37% of breast cancer survivors meet nutrition and physical activity guidelines, respectively [19].

A variety of short-term lifestyle interventions for breast cancer survivor populations have demonstrated successful weight loss [20,21]. While data on weight following the completion of these studies are limited, several studies have shown weight regain in the months following either the full intervention [22] or the intensive components of the intervention [23,24], prompting the question of whether measurable losses in body weight are sustainable for breast cancer survivors long-term.

The Lifestyle, Exercise, and Nutrition (LEAN) Study was a randomized-controlled weight-loss trial that compared the effect of in-person or telephone-based counseling versus usual care on changes in body composition, physical activity, diet, and serum biomarkers over six months in overweight or obese women with breast cancer [25]. The six-month trial led to a clinically meaningful 6% mean weight loss among the LEAN intervention group, compared to a 2% mean weight loss among the usual care group (*p* < 0.05) [25]. Given that the LEAN intervention provided a strong, clinically impactful short-term weight-loss benefit to breast cancer survivors, in the present analysis, we sought to determine the long-term impact of this intervention on weight change up to eight years post intervention. The primary aim of the current analysis was to evaluate long-term weight loss maintenance among breast cancer survivors enrolled in the LEAN Study. In an exploratory analysis, we also examined if weight change during the trial (weight loss, weight maintenance, weight gain) influenced weight change during long-term follow-up.

## 2. Materials and Methods

Women with BMI ≥ 25.0 kg/m^2^ diagnosed with Stage 0 to III breast cancer within five years prior to study enrollment were eligible for the LEAN Study (clinicaltrials.gov registration number NCT02109068). Eligible participants had completed chemotherapy and/or radiation therapy, were physically able to exercise, accessible by telephone, and able to read and communicate in English. Women were excluded if they were pregnant, intending to become pregnant within a year, had a history of stroke or myocardial infarction within six months, or had a severe uncontrolled mental illness. Participants were self-referred or recruited between June 2011 and December 2012 through the Breast Center at Smilow Cancer Hospital at Yale-New Haven Hospital and the Yale Cancer Center Survivorship Clinic; a total of 100 women were enrolled. The study was approved by the Yale School of Medicine Human Investigation Committee. The detailed protocol and primary results of the trial related to the intervention’s effect on 6-month change in body weight have been published previously [25].

Women were randomized to the LEAN intervention (either in-person or telephone-based counseling) or usual care group such that one-third of the participants were in each group. The weight loss intervention was centered around reduced caloric intake, increased physical activity, as well as behavioral therapy [25].

The intervention groups received 11 sessions of 30 min counseling led by a registered dietitian who was also a certified specialist in oncology nutrition, over the span of 6 months, either in person or via telephone, a breast cancer-specific healthy eating and exercise LEAN educational book, and a journal to guide counseling sessions. The in-person and telephone groups received the same lifestyle intervention. Participants received counseling sessions once per week throughout the first month, followed by every two weeks in the following two months, and then once per month in the final three months. The LEAN journal was used by participants to record all food and beverage intake, minutes of physical activity, and daily pedometer step counts, as well as their weight measured once a week on a scale provided by the study. Participants were provided with personalized energy intake goals based on baseline weight, such that they incurred an energy intake deficit of 500 kcal/day. The dietary fat goal was <25% total energy intake. Participants were encouraged to consume a plant-based diet and incorporate mindful eating practices alongside a home-based physical activity program with a goal of 150 moderate-intensity activity minutes per week and 10,000 steps per day [25].

The usual care group received one 30 min counseling session at the end of the six-month study period, in addition to the LEAN book and journal, American Institute for Cancer Research pamphlets on healthy eating and exercise, and referral to the Yale Cancer Center Survivorship Clinic, which offers a two-session weight management program [25].

Participant weights were measured by study staff in duplicate at baseline and the end of the six-month study period. Additional follow-up weight data assessed objectively via scales during patient visits at affiliated clinical sites through July 2019 was obtained retrospectively via patient electronic health records at Yale-New Haven Hospital.

The LEAN Study population comprised 100 breast cancer survivors; of those, 33 were randomized to usual care, 34 to intervention via telephone-based counseling, and 33 to intervention via in-person counseling. As there was no difference in weight loss between the two intervention groups, the telephone-based and in-person counseling groups were combined for this analysis [25]. After exclusion of participants who did not have six-month measured weights (end of LEAN Study), our analytic sample was 92 women (intervention = 60 (90%); usual care = 32 (97%)).

Participant weight trajectories were calculated between six months (end of LEAN Study) through July 2019 using the measured weight at the end of the LEAN Study and all available electronic health record data from thereon. Thus, up to 8 years of follow-up data were available.

Women’s baseline characteristics were summarized using descriptive statistics. A growth curve analysis using mixed effect modeling was performed to compare the rate of body weight change from the six-month endpoint of the original LEAN Study through the follow-up period. A random intercept effect was included to account for within-subject correlation among repeated assessments. The difference in the slopes of weight change over time was examined by including time and group interaction as a fixed effect. The slope represents yearly weight gain (if positive) or weight loss (if negative) in kilograms.

In addition, in an exploratory analysis, we categorized changes in body weight during the six-month LEAN Study period from baseline to six months into three levels: weight loss was defined as losing greater than 1% of body weight, weight gain was defined as gaining greater than 1% of body weight, and weight maintenance was defined as weight remaining within 1% of body weight. The weight trajectories were compared by randomization group and weight change category. All analyses were performed using SAS 9.4 (Cary, NC, USA). Statistical significance was set at *p* < 0.05, two-sided.

## 3. Results

### 3.1. Baseline Characteristics of Study Participants

At the start of the LEAN Study, the mean participant age was 58.8 years (SD = 7.3) with a mean BMI of 33.1 kg/m^2^ (SD = 6.6) (Table 1). The majority of participants were postmenopausal, identified as non-Hispanic whites, and had graduated from college. Among the 92 women in our analytic sample, the mean weight change over the six-month LEAN Study was significantly different between the intervention and usual care groups; on average, the intervention group lost 4.3 kg (5%), whereas the usual care group lost 1.6 kg (2%) (*p* = 0.009).

### 3.2. Post-Intervention Weight Change

The median number of weight data points per participant was similar between groups, 20 (intervention) and 18 (usual care). Women were followed up to eight years post intervention (mean = 5.9 SD = 1.9), and the mean years of follow-up was similar between groups, 6.0 (intervention) and 5.6 (usual care) (*p* = 0.38). In the post LEAN follow-up period, both groups had a decline in body weight over time (intervention = −0.20 kg or −0.2% per year, SE 0.07, *p* = 0.004; usual care = −0.32 kg or −0.4% per year; SE 0.10, *p* = 0.002) (Table 2) (Figure 1a,b). There was no statistically significant difference in the yearly mean rates of weight change between the intervention and usual care groups (*p* = 0.31).

### 3.3. Post-Intervention Weight Change by Weight Change during LEAN

The proportion of women who lost >1% body weight, gained >1% body weight, and maintained weight (>1% change) during the six-month LEAN Study was significantly different by study groups in our analytic sample (*p* = 0.0005) (Table 3). Considering weight-change categories during the LEAN Study, on average, women in the intervention group who lost weight during the LEAN Study, lost weight during follow-up (−0.09 kg per year), compared to those in the usual care group who had a non-significant gain in weight (0.20 kg per year) (group difference = −0.39 kg per year, *p* = 0.02) (Table 3). Women in both the usual care and intervention groups who gained weight during the six-month LEAN trial lost weight during follow-up; however, women in the usual care group had significantly greater weight loss (−1.32 kg per year) than those in the intervention group (−0.46 kg per year) (group difference = 0.85 kg per year, *p* = 0.002). Independent of the study group, women who maintained their weight during the study period did not experience significant weight change during follow-up (group difference = 0.08 kg per year, *p* = 0.82).

## 4. Discussion

Breast cancer survivors participating in the LEAN Study, regardless of randomization to usual care or intervention, avoided long-term weight gain after the lifestyle intervention. Given there was modest weight loss overall during long-term follow-up (average of 5.9 years), it was encouraging that the women in the intervention and usual care groups were able to maintain their original LEAN trial weight loss (6% weight loss for intervention versus 2% weight loss for usual care). Participation in a lifestyle intervention led to the prevention of weight gain over time and maintenance of clinically meaningful weight loss among breast cancer survivors randomized to the intervention. Our results provide evidence of the benefits of lifestyle programs; thus, clinicians should consider recommending and referring patients to cancer survivorship and weight management programs following a diagnosis of breast cancer.

Our finding that both intervention and usual care women in our study lost weight over time differs from what is typically seen in the general female population. For example, amongst adult women without obesity or chronic disease, a mean weight increase of 2.33 lbs. (1.06 kg) to 5.24 lbs. (2.38 kg) per 4 years has been shown [10], suggesting that, on average, adult women gain weight over time. Another long-term study following adult women for a mean of 26 years, found an average BMI increase of 3.7 kg/m^2^ and a mean weight change of 8.6 kg [26]. In this same study, baseline normal-weight women gained 2.4 kg more than obese women, and overweight women gained 3.3 kg more than obese women [26].

Although several lifestyle interventions similar to the LEAN intervention have demonstrated success in clinically meaningful weight loss among breast cancer survivors [20,21], several have observed weight regain in the months following either the full intervention [22] or the intensive components of the intervention [23,24]. A review of lifestyle interventions in female cancer survivors noted the challenge of maintaining participant motivation following the study conclusion and suggested that highly personalized approaches to weight loss may be more successful [27]. Thus far, the majority of existing studies have not reported long-term weight trajectories following the end of the intervention; therefore, additional research on this topic is vital. To our knowledge, this is the first study of long-term follow-up of a weight loss intervention in breast cancer survivors.

One possible explanation for our findings of modest weight loss during follow-up may be that all women enrolled in the LEAN Study had a BMI > 25 kg/m^2^. Not all of the data for the general population has been stratified by BMI at baseline, which may be an important predictor of weight change. As noted above, there are limited follow-up weight data from other weight-loss interventions in breast cancer survivors with overweight/obesity to compare our results. It is also possible that the breast cancer survivors in the LEAN Study may differ in terms of weight patterns from breast cancer survivors not enrolled in a lifestyle intervention, as women enrolled in LEAN were willing to participate in a randomized weight-loss trial of diet and exercise. Therefore, regardless of the randomization group, they may have had greater readiness to adopt healthy behaviors, including behaviors resulting in weight loss, following breast cancer treatment. Additionally, recent studies have indicated that premenopausal women [28,29,30,31] and those with lower BMI at diagnosis [32,33] appear to be at increased risk of post-diagnosis weight gain, and the original LEAN study did not target these populations. One additional explanation for the loss of weight during the long-term follow-up among the usual care study participants specifically may be that the one weight-loss counseling session and the study material that the women in this group received at the end of the six-month intervention was efficacious for modest weight loss.

Since the intervention group lost significantly more weight (6% weight loss) during the six-month LEAN Study than the usual care group (2% weight loss), we examined long-term weight change by weight change during the trial. In these analyses, we found women in the intervention group who lost >1% body weight during the trial continued to lose weight in the follow-up period, compared to women in the usual care group, who lost >1% body weight during the six-month study period and, in contrast, did not continue to lose weight during the follow-up period. Moreover, in both the intervention and usual care groups, women who maintained their weight during the trial continued to experience weight maintenance during follow-up. Lastly, women who gained weight during the LEAN Study lost weight during the follow-up independent of the intervention group. However, these exploratory analyses should be interpreted with caution, as we could not investigate the more traditional 5% weight change cut-points due to our small sample size.

Our long-term weight data are from electronic health records, and therefore, we had rates of weight change per year (e.g., slopes) rather than change at a defined follow-up time (e.g., one year or two years post intervention). Therefore, we could not assess predictors of long-term weight change in our population. However, as demonstrated in Table 1, the baseline characteristics are fairly balanced; thus, predictors of the trajectory of weight change over time are not significant for a comparison between groups. Additional research is needed to elucidate the frequency of post-diagnosis weight gain, maintenance, and loss amongst breast cancer survivors, including those with overweight/obesity at diagnosis and those receiving modern anti-cancer therapies. Large prospective studies of women diagnosed with breast cancer could also help us understand which women are at the greatest risk of post-diagnosis weight gain, and how this impacts prognosis.

This study has several limitations. Due to the use of electronic health records for the collection of weight histories during the follow-up period, we had a variable number of weight measurements and lengths of follow-up for participants. There may also be some measurement error in weight data in the electronic health record, but we tried to eliminate recording errors through extensive data cleaning and hand review of abstracted data, and the data in the records were objective from scales in clinical settings. We also could not assess other measures of body composition, such as lean mass versus fat mass during our long-term follow-up. These data could be useful in future studies to fully understand body composition in relation to cancer outcomes, as evidence suggests an increased risk of mortality among early breast cancer patients with sarcopenia [34]. Additionally, we were unable to assess if participants sought additional resources to promote weight loss during post-intervention follow-up. Our participants were from one institution, and the majority were college educated, non-Hispanic White, post-menopausal breast cancer survivors diagnosed with stage I breast cancer, meaning that these findings may not be applicable to all breast cancer survivors. For instance, African American communities have both higher rates of obesity and breast cancer mortality rates, compared to non-Hispanic white women; there are many factors that may contribute to these disparities, and further research to elucidate this relationship is necessary [35]. Important strengths of our study include the long-term follow-up period and multiple longitudinal-measured weights derived from the electronic health records, which eliminates the social desirability bias of self-reported weight.

## 5. Conclusions

In this sample of breast cancer survivors with overweight/obesity, we observed that overall women experienced modest long-term weight loss following the LEAN Study, even if randomized to usual care. Given our small size and the current lack of similar studies of long-term weight patterns following lifestyle interventions to which we could compare our results, additional research is necessary to understand long-term weight trajectories in breast cancer survivors with overweight/obesity both within and outside the context of lifestyle interventions.

## Figures and Tables

**Figure 1 nutrients-13-03265-f001:**
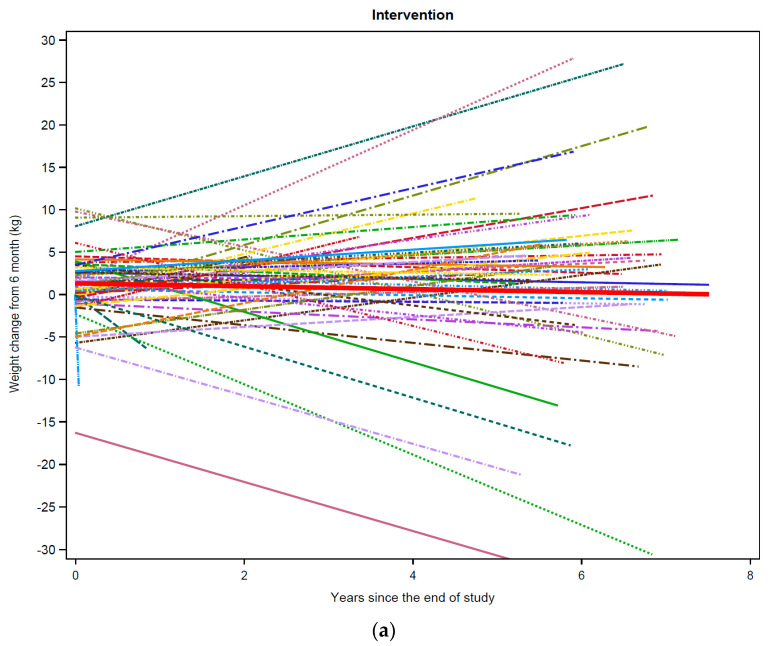
Weight trajectories by study group. Weight trajectories for each individual participant from LEAN Study completion through long-term follow-up (latest date July 2019) are shown: (**a**) intervention group (*n* = 60); (**b**) usual care (*n* = 32). The thick red line in (**a**) corresponds to the mean weight trajectory among intervention group participants, and the thick red line in (**b**) corresponds to the mean weight trajectory among usual care group participants.

**Table 1 nutrients-13-03265-t001:** Baseline characteristics of Lifestyle, Exercise and Nutrition (LEAN) Study participants with long-term weight data (*n* = 92).

Characteristic	Mean (SD) or *n* (%)			
	All *n* = 92	Intervention *n* = 60	Usual Care *n* = 32	*p*-Value
Age, years	58.8 (7.3)	59.4 (7.3)	57.6 (7.3)	0.28
BMI ^a^, kg/m^2^	33.1 (6.6)	32.7 (6.2)	33.9 (7.6)	0.42
College graduate	47 (51%)	33 (55%)	14 (44%)	0.30
Non-Hispanic white	84 (91%)	55 (92%)	29 (91%)	0.49
Postmenopausal	75 (82%)	50 (83%)	25 (78%)	0.54
Time from diagnosis to study enrollment, years	2.7 (1.8)	2.7 (1.5)	2.8 (2.2)	0.72
Post study follow-up time, years	5.9 (1.9)	6.0 (1.8)	5.6 (2.1)	0.38
Disease Stage				0.94
0	15 (16%)	9 (15%)	6 (19%)	
I	48 (52%)	31 (52%)	17 (53%)	
II	21 (23%)	15 (25%)	6 (19%)	
III	6 (7%)	4 (7%)	2 (6%)	
Unknown	2 (2%)	1 (1%)	1 (3%)	
Treatment after surgery				0.67
None	14 (15%)	8 (13%)	6 (19%)	
Radiation only	34 (37%)	21 (35%)	13 (41%)	
Chemotherapy only	17 (18%)	13 (22%)	4 (13%)	
Radiation and Chemotherapy	27 (29%)	18 (30%)	9 (28%)	
Weight (kg)				
Baseline	87.5 (18.1)	86.1 (16.8)	90.4 (20.3)	0.27
Six-month	84.4 (19.3)	82.4 (18.0)	88.3 (21.2)	0.16
Weight change within study period (kg)	−3.4 (5.3)	−4.3(5.7)	−1.6 (3.7)	0.009

^a^ BMI, body mass index.

**Table 2 nutrients-13-03265-t002:** Mean weight trajectories during the follow-up period by group.

	Yearly Mean Rate of Weight Change (kg)	SE ^b^	95% CI ^c^	*p*-Value
Intervention ^a^ (*n* = 60)	−0.20	0.07	[−0.06, −0.33]	0.004
Usual care (*n* = 32)	−0.32	0.10	[−0.12, −0.53]	0.002

^a^ By-randomization group *p*-value was 0.31. ^b^ SE, standard error. ^c^ CI, confidence interval.

**Table 3 nutrients-13-03265-t003:** Post-study weight change stratified by weight changes at the end of six-month study.

	Intervention (*n* = 60)	Usual Care (*n* = 32)	Chi-Square *p*-Value		
**Overall Weight Change**			0.0005		
Weight loss (>1% loss)	48 (80%)	16 (50%)			
Weight gain (>1% gain)	8 (13%)	7 (22%)			
Weight maintenance (>1% change)	4 (7%)	9 (28%)			
	**Weight Change (kg/yr.)**	**SE**	**Lower**	**Upper**	***p*-Value**
**Intervention**					
Weight loss during LEAN	−0.09	0.04	−0.15	−0.02	0.02
Weight gain during LEAN	−0.46	0.18	−0.82	−0.10	0.01
Weight maintenance during LEAN	−0.07	0.25	−0.57	0.42	0.78
**Usual Care**					
Weight loss during LEAN	0.20	0.15	−0.08	0.49	0.16
Weight gain during LEAN	−1.32	0.20	−1.71	−0.92	<0.0001
Weight maintenance during LEAN	−0.15	0.25	−0.64	0.34	0.54
**Intervention vs. Usual=Care Group Comparison**					
Weight loss: intervention vs. usual care	−0.39	0.16	−0.71	−0.06	0.02
Weight gain: intervention vs. usual care	0.85	0.27	0.32	1.39	0.002
Weight maintenance: intervention vs. usual care	0.08	0.35	−0.62	0.78	0.82

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
