# Peer review of "Effect of the Lifestyle, Exercise, and Nutrition (LEAN) Study on Long-Term Weight Loss Maintenance in Women with Breast Cancer"

_nutrients, 2021, doi:10.3390/nu13093265_

Round 1
Reviewer 1 Report
Dear Authors,
Please accept congratulations on this research and obtained results.
The value of the introduction/discussion would be better after including a short description of current US guidelines on obesity treatment including recommendations for pharmacotherapy.
Please also indicate (if applicable) how current US recommendations address obesity tretment in beast cancer survivors (BCS) and how the LEAN study refers to them?
How the clinical situation of BCS change the possibilities of applying these recommendations to this clinical group?
What practical, clinical implications would you draft/formulate based on your results for clinical oncologists/physicians taking medical care of BCS (IF ANY)?
Please address also racial/ethnic differences regarding the risk of breast cancer as related to obesity. How this question regards your research group?
Author Response
Response to Reviewers
We thank the reviewers for the comments. Our detailed responses are below and changes to the manuscript are shown with track changes.
Reviewer #1
1.The value of the introduction/discussion would be better after including a short description of current US guidelines on obesity treatment including recommendations for pharmacotherapy.
While we understand that pharmacotherapy is one option for obesity treatment, this falls outside the scope of our lifestyle-based intervention as well as our research expertise in nutrition and physical activity in relation to obesity. In addition, since our work is done among cancer survivors, it is likely that any pharmacotherapy for obesity would need to be carefully considered on a patient-by-patient basis for this type of population and therefore, we do not feel it would be possible to discuss all of the nuances of this type of treatment in our paper.
- Please also indicate (if applicable) how current US recommendations address obesity treatment in breast cancer survivors (BCS) and how the LEAN study refers to them?
We agree this would be helpful to the readers and we have now provided a brief synopsis of the 2016 ACS and ASCO breast cancer survivorship care guidelines in regard to nutrition, physical activity, and weight management.
- How the clinical situation of BCS change the possibilities of applying these recommendations to this clinical group?
We have now included reference to guidelines that are specific to breast cancer survivors and will note any ways in which breast cancer survivorship may impact application of guidelines regarding nutrition, physical activity, and weight management.
- What practical, clinical implications would you draft/formulate based on your results for clinical oncologists/physicians taking medical care of BCS (IF ANY)?
We have added a statement in the discussion about integrating lifestyle counseling into clinical care.
- Please address also racial/ethnic differences regarding the risk of breast cancer as related to obesity. How this question regards your research group?
We have noted in our discussion that our study is limited by our study population being mainly college educated, non-Hispanic White, post-menopausal breast cancer survivors, and that our results may not be applicable to all patients. We have updated the discussion to raise how race/ethnicity could play an important role for risk of breast cancer and obesity.
Reviewer 2 Report
The authors aim to evaluate long-term weight loss maintenance in breast cancer survivors with overweight/obesity enrolled in a six-month randomized controlled trial.
I have a few comments:
- According to the authors’ guidelines, the abstract should be a total of about 200 words maximum. The abstract should be a single paragraph and should follow the style of structured abstracts. Please, correct this point.
- More detail should be provided on study design and endpoints.
- The exact nature of the intervention isn't clear. In the methods section, we expect to see a full description of the approach, with enough details that another research group could reproduce the intervention.
- The numbers in Table 1 don't match (e.g. 14 + 34 + 17 + 27 is not 92, as well as 9 + 31 + 15 + 4 is not 60).
- It seems that the sentence “Breast cancer survivors participating in the LEAN Study, regardless of randomization to usual care or intervention, avoided long-term weight gain after the lifestyle intervention with evidence of weight loss over follow-up” is repeated; please, correct this point.
- They should discuss the potential biological mechanisms linking the relationship between overweight/obesity and breast cancer biology and prognosis.
- Despite the study focus on weight change, the authors have not considered the impact of the lean mass/fat mass in the discussion.
- The language is OK although some improvement is needed in terms of grammar and sentence construction. The discussion section is somewhat messy.
- I couldn't file clinical trials register for this investigation on clinicaltrials.gov. Was this investigation registered?
- The authors should integrate the reference list. I suggest quoting the following references:
- Chlebowski RT, et al. Weight loss in breast cancer patient management. J Clin Oncol. 2002;20:1128–1143.
- Chlebowski R.T., Reeves M.M. Weight loss randomized intervention trials in female cancer survivors. J Clin Oncol. 2016;34:4238–4248.
- Trestini I, et al. Evidence-based tailored nutrition educational intervention improves adherence to dietary guidelines, anthropometric measures and serum metabolic biomarkers in early-stage breast cancer patients: A prospective interventional study. 2021 Aug 21;60:6-14.
Author Response
Response to Reviewers
We thank the reviewers for the comments. Our detailed responses are below and changes to the manuscript are shown with track changes.
Reviewer #2
- According to the authors’ guidelines, the abstract should be a total of about 200 words maximum. The abstract should be a single paragraph and should follow the style of structured abstracts. Please, correct this point.
We have reduced the abstract word count to match the guidelines.
- More detail should be provided on study design and endpoints.
Since the primary paper on the LEAN intervention has been published these are described in full elsewhere. We have added brief additional details regarding study design and endpoints in the methods section.
- The exact nature of the intervention isn't clear. In the methods section, we expect to see a full description of the approach, with enough details that another research group could reproduce the intervention.
This has been revised and further detail has been added.
- The numbers in Table 1 don't match (e.g., 14 + 34 + 17 + 27 is not 92, as well as 9 + 31 + 15 + 4 is not 60).
This has been rectified; an additional category regarding staging has been added.
- It seems that the sentence “Breast cancer survivors participating in the LEAN Study, regardless of randomization to usual care or intervention, avoided long-term weight gain after the lifestyle intervention with evidence of weight loss over follow-up” is repeated; please, correct this point.
This has been rectified; the duplicate sentence has been removed.
- They should discuss the potential biological mechanisms linking the relationship between overweight/obesity and breast cancer biology and prognosis.
We have now included proposed mechanisms underlying the relationship between overweight and breast cancer to the introduction.
- Despite the study focus on weight change, the authors have not considered the impact of the lean mass/fat mass in the discussion.
We did not have longitudinal measures of lean mass/fat mass during this long-term follow-up so this was not focus of the research. However, we have incorporated the relevance of lean mass and fat mass in the discussion.
- The language is OK although some improvement is needed in terms of grammar and sentence construction. The discussion section is somewhat messy.
The manuscript has been reviewed for grammar and readability.
- I couldn't file clinical trials register for this investigation on clinicaltrials.gov. Was this investigation registered?
The LEAN study was registered when the original trial was conducted and the clinicaltrials.gov number is now included in the methods section.
- The authors should integrate the reference list. I suggest quoting the following references:
- Chlebowski RT, et al. Weight loss in breast cancer patient management. J Clin Oncol. 2002;20:1128–1143.
- Chlebowski R.T., Reeves M.M. Weight loss randomized intervention trials in female cancer survivors. J Clin Oncol. 2016;34:4238–4248.
- Trestini I, et al. Evidence-based tailored nutrition educational intervention improves adherence to dietary guidelines, anthropometric measures and serum metabolic biomarkers in early-stage breast cancer patients: A prospective interventional study. 2021 Aug 21;60:6-14.
We have reviewed these references and incorporated as appropriate to our discussion section.